# The Piston Elastic: A Novel Device for Treating Entrapped Ectopic Permanent Molars

**DOI:** 10.3390/children8080652

**Published:** 2021-07-28

**Authors:** Ik-Hwan Kim, Chung-Min Kang, Je Seon Song, Jaeho Lee, Hyung-Jun Choi, Seong-Oh Kim

**Affiliations:** Department of Pediatric Dentistry, College of Dentistry, Yonsei University, Seoul 03722, Korea; kih86007@yuhs.ac (I.-H.K.); kangcm@yuhs.ac (C.-M.K.); songjs@yuhs.ac (J.S.S.); leejh@yuhs.ac (J.L.); choihj88@yuhs.ac (H.-J.C.)

**Keywords:** ectopic eruption, permanent molar, tooth eruption, impaction

## Abstract

Ectopic eruption of the permanent molar may absorb the distal root of the primary second molar and may result in a decreased arch length or delayed eruption of the permanent tooth, requiring timely treatment. Therefore, we devised an effective and convenient method to unlock the entrapped tooth using a novel device called a “piston-elastic”. This case report aims to explain the design and clinical application of this piston-elastic and to describe successful cases. Three patients (aged 6, 13, and 16 years) with ectopically erupted maxillary and mandibular molars, respectively, were treated with a piston-elastic. It was bound to the locked molar to improve the eruption path. After a certain time period, the repulsive force pushed the surface of the adjacent tooth, improving the eruption path of the entrapped tooth. The piston-elastic is a novel device that simply and effectively changes the direction of eruption of ectopically entrapped molars. As it can be manufactured and attached to the chair side, impression acquisition on a model cast and laboratory procedures are unnecessary. Compared to existing methods, the piston-elastic can be easily produced and delivered, causes little irritation, and is inexpensive.

## 1. Introduction

Ectopic eruption refers to a phenomenon in which a tooth erupts from an abnormal position [1,2]. The prevalence of ectopic eruption of the first permanent molar is 2–6%, and more frequently occurs in the maxilla than in the mandible, whereas that of the second permanent molar is 0.06–1.7%, which more frequently occurs in the mandible than in the maxilla [3,4].

In particular, ectopic eruption of the first molar often results in the absorption of the distal root of the primary second molar, i.e., the adjacent tooth. Without appropriate treatment of the partially erupted locked first molar, increased mobility or premature loss of the primary second molar and difficulty in oral hygiene management around the ectopic-erupted tooth may occur. Consequently, the space for permanent tooth eruption may be lost, and dental caries and localized gingivitis may develop [5].

Ectopic eruption of a molar is easily detected in regular check-ups and requires early intervention to prevent sequelae. Ref. [6] Several treatment devices, such as brass wire, elastic separators, Kesling spring, K-Loop, Rect-spring, Halterman appliances, and Humphrey appliances, have already been used [7,8,9,10]. Herein, we introduce a novel device known as a “piston-elastic” that can effectively improve molar ectopic eruption. This report describes the improvement of ectopic permanent molars in cases treated using piston-elastics, and provides details on the characteristics of this device.

The smallest #7 metal tube (0.018 × 0.022 slot) is used to make a piston-elastic, and a straight 0.016 × 0.022 stainless steel wire is prepared. A rectangular wire is necessary to prevent the swinging motion. The basic concept is to create a key and keyhole, made of the rectangular wire and tube, respectively. The wire is bent to create the key shape. Figure 1A shows the right end, indicating the “stop” side, which will limit the wire length preventing its loss, whereas the left end is the “hook” used to hang the elastic. The angle between the straight part of the wire and the hook should be <90° to prevent slipping of the elastic.

The following clinical circumstances should be considered: first, to use this device, the occlusal surface of the locked tooth must be sufficiently exposed because the device will be attached to the exposed occlusal surface. If the amount of exposure on the occlusal surface is insufficient, operculectomy or window opening surgery could be performed. Otherwise, the device could be attached six months later after there is an increase in the area of the occlusal surface.

Second, the elastic must be pre-installed on the device before attaching it to the tooth. Since the device is very small in size and is going to be attached to a limited visual access area on the back of the mouth, it would be difficult to install a new elastic after the attachment procedure. The elastic is pre-placed between the base of the tubed bracket and the pre-formed wire. It should not be activated by engaging the elastic to the hook before the attachment. If it is inserted into the hook of the wire in advance, this premature activation interrupts the proper location and fixation of the device. Therefore, inserting the elastic into the hook is performed after the device is firmly attached. After pre-installing the elastic, the “stop” side of the device should be as short as possible (Figure 1A). The shorter the stop side length, the more you can push.

Third, a conventional light-cured orthodontic bracket bonding system is used to attach this device. After acid etching the exposed occlusal surface of the tooth, a bonding agent is applied and cured by light. Subsequently, the adhesive paste is placed over the base of the device on which the elastic is passively pre-installed, and the device is pressed against the occlusal surface of the tooth (Figure 1A). At this time, the “stop” side of the steel wire should be placed on a front side, slightly touching the distal surface of the adjacent molar. With this arrangement, the hook of the wire can be seen on the posterior side. This state shows the inverted orientation of the bracket tube so that the power arm is located in a distal direction, which is correct for the piston-elastic. The preformed wire part should be assessed such that both ends of the wire do not protrude too much, thereby preventing the impingement of the surrounding soft tissues. Once correctly positioned, the excess resin around it is carefully removed. The excess resin should not touch the steel wire part. If excess resin adheres to the steel wire and polymerized, it may interfere with the normal function of the device. Finally, the adhesive resin paste is polymerized by a light-curing unit. If the patient’s head moves when you shake the attached device with dental forceps, it is considered to be properly attached.

Fourth, activate the device when the device is firmly attached to the tooth. The pre-installed elastic on the device is pulled and hung on the distal hook of the steel wire. Then, the elastic is passed through behind the power arm such that it does not get cut during mastication. Since the elastic has its own elastic force, it can provide the distalizing force of the locked molar. A connection between the hook of the steel wire and the power arm of the bracket is not recommended because this length is too short for the proper action of the elastic, and the device may not produce enough force. Increasing the length of the wire to allow for sufficient elastic force is not recommended, as this will result in mucosal impingement as the device becomes longer.

Finally, the elastic can be replaced during recall check-ups, which can deliver prolonged distalizing force. The elastic can be used with a power chain, O-ring, or zinc string. Regular check-ups are recommended at three to four-week intervals, accompanied by radiographic examination and elastic changes

## 2. Case Report

Case 1: A six-year-old girl was referred from a local clinic to the Department of Pediatric Dentistry at our University with a complaint of ectopically erupting permanent maxillary right and left first molars. Her medical and dental histories were normal. Radiographic examination revealed a mesioangulated permanent maxillary first molar with partial root resorption of the primary second molar. A piston-elastic device was attached to both of the maxillary first molars. The patient performed regular check-ups every three to four weeks, and the elastic was replaced. Within three months, the eruption direction of the teeth was confirmed to be improved (Figure 2).

Case 2: A 12-year-old healthy girl without any history of dental trauma or systemic disease visited the same clinic. As mesially angulated impaction of the lower second molars was found bilaterally, and piston-elastic devices were attached. This patient also had regular check-ups every three to four weeks, and the elastic was replaced. After three months, directions of the lower second molars improved, showing a full eruption reaching occlusal contact (Figure 3). 

Case 3: A 13-year-old healthy girl was receiving orthodontic treatment at our clinic. Mesially angulated ectopic impaction of the right second molar was observed during orthodontic treatment. As the buccal surface area was not enough for bracket attachment, a piston-elastic device was attached to the exposed occlusal surface. After attaching the device, regular check-ups were performed four weeks later, and the elastic was replaced only once. At two months post-attachment, a definite change in the eruption path was observed, and the tooth had successfully erupted, completing the orthodontic treatment (Figure 4).

## 3. Discussion

The cause of ectopic eruption has not been accurately identified yet; however, it has been speculated to be caused by various factors [2,11,12,13,14]. Ectopic eruption of the permanent molar appears mostly as a type of mesial tilting [15]. In particular, ectopic eruption of the first molar is accompanied by distal root resorption of the primary second molar. Ectopic eruption of the permanent molar does not cause serious symptoms such as pain, discomfort, or infection, nor does it cause esthetic problems such as ectopic eruption of the anterior teeth. However, the permanent molar plays a major role in mastication and occlusion and must be treated.

Several methods have been used to treat ectopic eruption of molars. Choosing an appropriate treatment method according to tooth condition is very important. The use of an elastic separator is one of the simplest ways to separate the locked teeth. A brass wire and separating springs can be easily made at the chairside [16,17]. However, local anesthesia is required to insert a wire in the interproximal area. These methods are only suitable when the amount of locking is very small, owing to the mechanism of the intrusive force. In addition, they have limitations in regaining sufficient space and are ineffective. 

Most ectopic eruptions of molars can be improved through distal movement [15]. In more severe cases, a Halterman and Humphrey appliance, among others, can be used [8,18]. However, these conventional devices require impression-taking to manufacture a model cast and have a prolonged laboratory processing time, resulting in an increased cost and treatment duration. Furthermore, patients feel discomfort when these devices are set in the oral cavity due to the volume of the device, which is disadvantageous in oral hygiene management. As a result, dental plaque accumulation may cause dental caries or deterioration in periodontal health of the adjacent teeth.

Spontaneous improvements are not always observed for ectopically erupting molars. If the ectopic entrapment of the first molar was found before age seven, periodic re-evaluation might be performed every six months to determine whether there is a spontaneous improvement. After seven years of age, if the eruption path of the molars is poor, treatment with this device may be attempted immediately.

When we observe the entrapment of the upper molar, the typical locked part is the mesiobuccal cusp. (Figure 1A) This happens due to the rotation of the tooth. As the palatal root is the largest in upper molars, it rotates around this root when it moves underneath the adjacent tooth. Therefore, when we mount the piston-elastic on the typically locked upper molars, de-rotation around the palatal root is generally observed (Figure 1B) On the other hand, in cases of the entrapped mandibular molars, these tend to improve by distal tilting rather than rotation.

However, unlike other devices, the piston-elastic is attached to the occlusal surface, so it has a limitation in that it may fall out during mastication or as the eruption path of the tooth may be improved. In general, patients spit out the device that has been dropped out, or it can be swallowed. Therefore, caution is required. If the patient swallows the device, it is usually excreted within a few days. However, it can cause pneumonia in cases of aspiration. Therefore, it is necessary to closely monitor whether there is a persistent cough when the device drops out. Additionally, careful attention is needed in patients with abnormal movements such as bruxism. If it drops out repeatedly, or abnormal habits like bruxism or clenching are observed, the addition of the bilateral posterior bite blocks should be considered.

Overall, our novel “piston-elastic” device can overcome the disadvantages of existing devices and can ensure effective treatment. This device can be easily manufactured at the chairside with only a second molar standard tube bracket, rectangular stainless-steel wire, and separating elastic. Furthermore, ready-made forms can also be prepared. In addition, it can be easily delivered, allows tooth movement by several millimeters, and causes less discomfort due to its small size compared to existing devices. A piston-elastic may not be the first treatment choice for entrapped teeth, but it may be considered in severe cases before choosing Halterman or Humphrey appliances such as the Rect-spring [10].

## 4. Conclusions

The eruption path of the entrapped tooth should be improved in a timely manner. Compared to conventional devices, piston-elastic devices are much more efficient in design, less complex, do not require local anesthesia, impression taking, and laboratory procedures, or cause gingival irritation. A piston elastic can be effectively utilized even in cases with locking of several millimeters or more. They can replace conventional devices in particular cases. The piston-elastic is not the first-choice device for ectopic erupted molars; however, in cautious case selection, it can be another useful device.

## Figures and Tables

**Figure 1 children-08-00652-f001:**
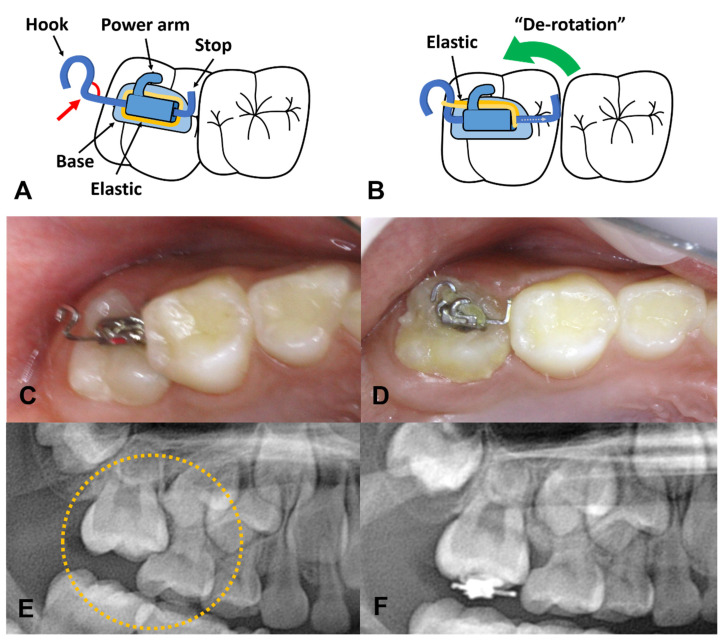
Design of the piston-elastic and its action. (**A**,**C**,**E**) Piston-elastic is bonded on occlusal surface of locked molar with the hook located on distal side. The elastic is placed such as in the figure. In order to prevent the slipping off of an elastic, the angle between the straight part of the wire and the hook should be less than 90 degrees (red arrow). (**B**,**D**,**F**) By the mechanism of piston-elastic, the ectopic molar gradually moves toward the distal side.

**Figure 2 children-08-00652-f002:**
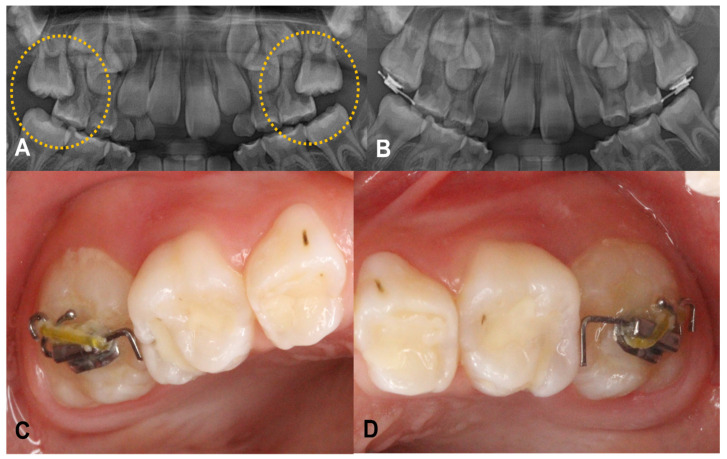
A 6-year-old girl showed ectopically erupting permanent maxillary right and left first molar. (**A**) Severe resorptions of the distal root of maxillary primary molars on both sides were observed on panoramic view. (**B**) Result of piston-elastic, 3 months after the attachment. (**C**,**D**) Intraoral photograph of the device.

**Figure 3 children-08-00652-f003:**
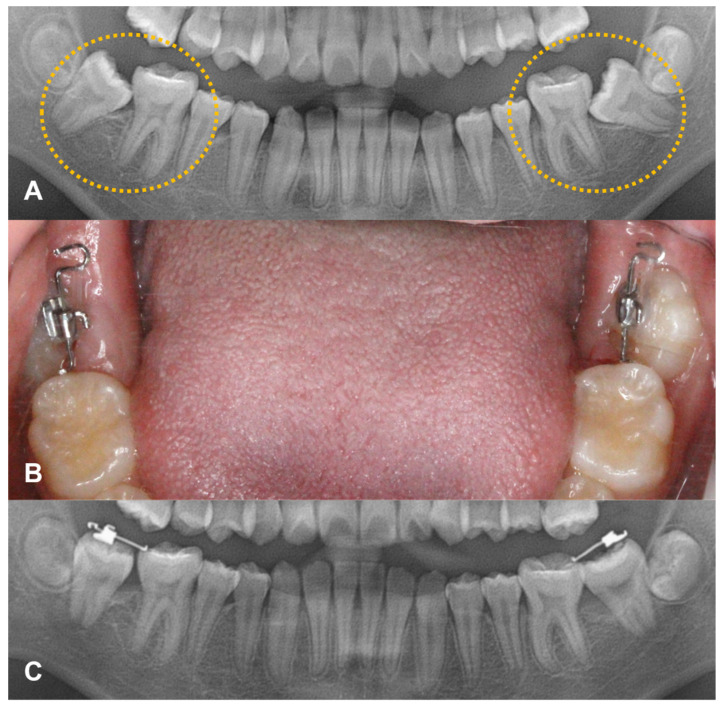
Case of second molar uprighting. (**A**) A 12-year-old-girl shows the mesially angulated impaction of lower second molars bilaterally. (**B**) Piston-elastic devices were attached (**C**) After three months, the directions of lower second molars were improved.

**Figure 4 children-08-00652-f004:**
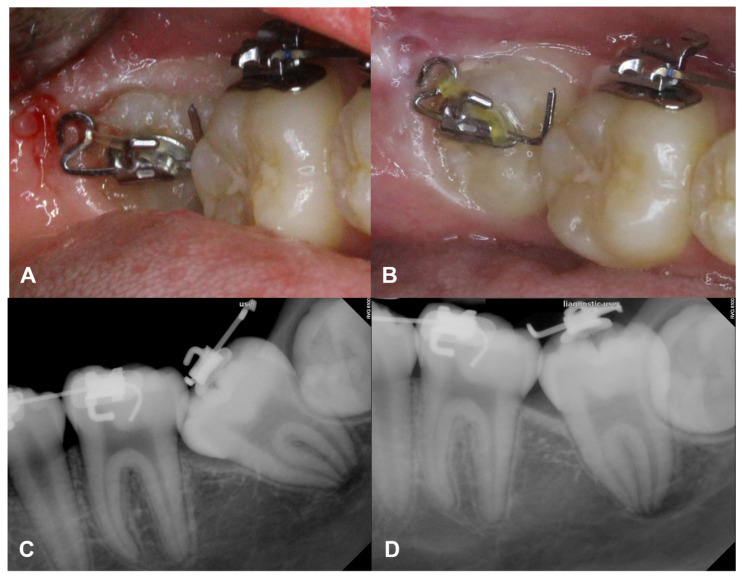
Piston-elastic during the orthodontic treatment. (**A**,**C**) Mesially angulated ectopic impaction of the right second molar was found during orthodontic treatment in a 13-year-old healthy girl. (**B**) As it showed the limited buccal surface for the bracket attachment, a piston-elastic device was attached on the exposed occlusal surface. (**D**) After 2 months of the attachment, a definite change in the eruption path was observed.

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
