# Peer review of "The Piston Elastic: A Novel Device for Treating Entrapped Ectopic Permanent Molars"

_children, 2021, doi:10.3390/children8080652_

Round 1

Reviewer 1 Report

Without being a very innovative theme, it is clinically relevant due to the high frequency of occurrence; the simplified approach presented may represent an interesting alternative. Please consider that bibliography can be updated with more recent references.

Author Response

We sincerely thank you for reviewing the insufficient thesis.

The content was supplemented for the part you mentioned, and the bibliography of the underlying key journal was maintained, but the latest reference was added.

Thank you very much.

Reviewer 2 Report

The authors present a self-developed orthodontic tool to correct mechanically caused eruption problems of the first molars. Details of the appliance are prescribed together with 4 example cases. The text is well written and easy to read. This reviewer would like to suggest supplementing the paper with the aspect of indication for treatment. It is known from the clinic as well as from the literature that some of the retained molars erupt spontaneously after a while (“self correction“ or „jump type“ versus „reversal type“ or „hold type“).

See as examples:

  1. Hennessy J, Al-Awadhi EA, Dwyer LO, Leith R. Treatment of ectopic first permanent molar teeth. Dent Update. 2012;39:656-8, 660-1.
  2. Chen X, Huo Y, Peng Y, Zhang Q, Zou J. Ectopic eruption of the first permanent molar: Predictive factors for irreversible outcome. Am J Orthod Dentofacial Orthop. 2021;159:e169-e177.

Author Response

Thank you very much for reviewing and guiding the insufficient thesis.

Regarding the parts you mentioned, I was able to supplement the missing parts and learn a lot through the process.

Once again, please evaluate the revised manuscript.

Thank you sincerely.